# Application of High-Throughput Sequencing for Comprehensive Virome Profiling in Grapevines Shows Yellows in Iran

**DOI:** 10.3390/v16020204

**Published:** 2024-01-29

**Authors:** Zahra Gholampour, Mohammad Zakiaghl, Elisa Asquini, Mirko Moser, Valeria Gualandri, Mohsen Mehrvar, Azeddine Si-Ammour

**Affiliations:** 1Department of Plant Pathology, College of Agriculture, Ferdowsi University of Mashhad, Mashhad 9177948978, Iran; zahra.gholampour@mail.um.ac.ir (Z.G.); mehrvar@um.ac.ir (M.M.); 2Research and Innovation Center, Fondazione Edmund Mach, 38098 San Michele All’Adige, Italy; elisa.asquini@fmach.it (E.A.); mirko.moser@fmach.it (M.M.); valeria.gualandri@fmach.it (V.G.)

**Keywords:** virome, sRNAome, virus mixed infection, *Vitis vinifera*, Iran

## Abstract

A comprehensive study on the whole spectrum of viruses and viroids in five Iranian grapevine cultivars was carried out using sRNA libraries prepared from phloem tissue. A comparison of two approaches to virus detection from sRNAome data indicated a significant difference in the results and performance of the aligners in viral genome reconstruction. The results showed a complex virome in terms of viral composition, abundance, and richness. Thirteen viruses and viroids were identified in five Iranian grapevine cultivars, among which the grapevine red blotch virus and grapevine satellite virus were detected for the first time in Iranian vineyards. Grapevine leafroll-associated virus 1 (GLRaV1) and grapevine fanleaf virus (GFLV) were highly dominant in the virome. However, their frequency and abundance were somewhat different among grapevine cultivars. The results revealed a mixed infection of GLRaV1/grapevine yellow speckle viroid 1 (GYSVd1) and GFLV/GYSVd1 in grapevines that exhibited yellows and vein banding. We also propose a threshold of 14% of complete reconstruction as an appropriate threshold for detection of grapevine viruses that can be used as indicators for reliable grapevine virome profiling or in quarantine stations and certification programs.

## 1. Introduction

Grapevine (*Vitis vinifera* L.) is an economically important plant worldwide [1]. As a woody plant that reproduces clonally, it is susceptible to single or mixed infections with viruses and viroids [2]. To date, more than 86 viruses and five viroids have been reported from vineyards worldwide [3]. Many of these viruses (e.g., arabis mosaic virus, ArMV; grapevine fanleaf virus, GFLV; grapevine leaf roll-associated virus, GLRaV; tomato ringspot virus, ToRSV) are prevalent worldwide, and some others (e.g., artichoke Italian latent virus, AILV; grapevine Anatolian ringspot virus, GARSV; and grapevine Bulgarian latent virus, GBLV) are restricted to a certain geographic area [3].

This large number of viral infections in a plant indicates a more comprehensive view of pathogen-host interaction as a dynamic micro-ecosystem instead of a single pathosystem. This change in perspective has led to increased knowledge of disease epidemiology, prediction of potential threats, and the development of effective control strategies for disease management [4,5]. Several studies have been conducted to determine the population dynamics of viruses in a grapevine or a vineyard [3,6]. Sometimes, grapevine viruses do not follow the classical concept of “one pathogen-one disease”, so the interaction of more than one viral agent leads to the development of disease symptoms [2,4,7]. The synergistic or antagonistic interaction of viruses in co-infections can influence symptoms, viral concentration, and ultimately disease severity [8,9].

The use of specific virus detection methods, e.g., serology or PCR, is laborious and time-consuming, especially in a plant harboring a large number of viruses, viroids, and phytoplasmas, and cannot provide a clear overview of the plant’s virome [2]. Over the past decade, a new era in virology has begun with the employment of high-throughput sequencing (HTS) [10]. It does not require prior knowledge of the viruses, so the identification of viruses/viroids is achieved only by determining the sequence of small RNAs [11]. HTS technologies also diagnose co-infections of viruses in plants, which are much more common than single infections [12]. Viruses/viroids are targeted by the Dicer-mediated silencing machinery, resulting in the accumulation of virus/viroid-derived small RNAs (sRNA) of 21–24 nucleotides in length in infected host cells [13]. Since sRNAs are derived from the entire viral genome, their analysis enables the reconstruction of the complete genome of infected viruses and viroids by sequencing and de novo assembly of sRNAs [14,15,16,17]. HTS generates a large amount of sequencing data from each sample and requires appropriate data analysis. Various bioinformatics tools have been developed to examine the data from HTS, and several pipelines have been developed for plant virus detection [18]. Many of these software programs have been developed on an open-source Unix platform with pipelines to perform analyses; commercial analysis packages such as CLC Genomic Workbench or Geneious are also available.

The grape industry in Iran is constantly expanding; from more than 170.3 thousand hectares of vineyards, approximately 2.4 million tons of table grapes were produced in 2021. Khorasan-Razavi province in northeastern Iran is the third largest grape producer in Iran (Statistical Yearbook of Jihad-Agriculture, 2022 [19]. More than a hundred *V. vinifera* varieties are grown in Iran, but five varieties (including cv. Peykani, cv. Askari-Bidane, cv. Rezghi, cv. Sahebi, and cv. Fakhri) are the predominant grapevine varieties in Khorasan-Razavi.

Despite the vast areas of vineyards in Iran, few studies have been conducted to identify grapevine viruses; moreover, all the studies conducted were only focused on one/few viruses or viroids detected by specific detection methods [20,21,22].

In northeastern Iran, yellows and a progressive decline in vines are observed in many vineyards. The infected grapevines express moderate to severe yellows, vigor reduction, and decline. Reports indicate the wide prevalence of GFLV in vineyards [23], but in many cases, RT-PCR is not able to detect GFLV in infected grapevines [24]. Also, the phytoplasmas flavescence doree and bois noir, which are reported as causal agents of grapevine yellows [25], were not detected in the infected grapevines. Therefore, sRNA libraries were prepared from the symptomatic grapevines to identify the virome by NGS. The objective of this study is to (1) investigate and compare the virus populations in five Iranian commercial V. vinifera cultivars and (2) determine the relationship between the identified viruses and the yellowing and decline syndrome of grapevines in vineyards in northeastern Iran. The present study is the first report of grapevine virome in Iran using HTS data from commercial Iranian grapevine cultivars using phloem tissue.

## 2. Materials and Methods

### 2.1. Plant Material

Plant materials were collected from a severely diseased vineyard in northeastern Iran. Based on symptoms, six vines belonging to five Iranian grape cultivars were collected from vineyards in June 2018 (Table 1 and Figure 1).

### 2.2. Library Preparation and Sequencing

Total RNA was extracted from one gram of petioles using an extraction protocol developed by Carra et al. [26]. The low molecular weight RNA fraction (LMW-RNA) was participated using polyethylene glycol 6000; subsequently, small RNAs were separated in a 7% polyacrylamide gel by electrophoresis and recovered from the gel [27].

Small RNA libraries from the six grapevines were constructed using TruSeq Small RNA Sample Prep Kits (Illumina, San Diego, CA, USA). The integrity and concentration of the libraries were assessed using KAPA Library Quantification Kits (Roche, Basel, Switzerland) and an Agilent DNA 1000 Kit. Finally, the libraries were sequenced in Illumina Novaseq 6000 by Novogene (Cambridge, UK).

### 2.3. Analysis of Sequencing Data and Identification of Viruses

The quality of the reads was checked using FastQC [28], and low-quality reads and adaptors were removed using Cutadapt 3.0 [29]. Two approaches were used to detect viruses in the libraries (Figure 2).

First, short reads from each library were assembled using a velvet assembler [30] with a k-mer in the range of 13 to 25. Contigs of each k-mer were individually subjected to BLASTN and BLASTX against the nr database (http://www.ncbi.nlm.nih.gov/, accessed on 25 May 2022) with an e-value threshold of 10^−6^ to identify known viruses using CLC Genomics Workbench 12 (CLC Bio, Aarhus, Denmark) software. Contigs identified in both BLASTN and BLASTX were used for subsequent analyses.

In the second strategy, the automated VirusDetect v1.6 pipeline [31] was used to discover viruses found in the virome of grapevine.

Contigs mapped to the genomes of *V. vinifera*, bacteria, and fungi were omitted for further analysis. Based on the results from BLAST, a list of viruses and viroids found in the libraries was compiled using most homologs in GenBank (Appendix A). 

To compare the performance of the different assemblers in reconstructing the viral genome, the short reads were mapped to the reference genomes using the CLC Genomics Workbench with a mismatch tolerance of two or the Bowtie aligner [32] implemented in the UGENE package [33] with the following parameters: Mischmatch cost = 2 (cost of a mismatch between the read sequence and the reference sequence); insertion cost = 3 (cost of an insertion in the read sequence causing a gap in the reference sequence); and deletion cost = 3 (cost of a gap in the read sequence) using global alignment and randomly aligned read sequences. In addition, the assignment of reads to the reference genome was performed automatically in VirusDetect v1.7 using the v229 virus reference database. 

### 2.4. Confirmation of NGS Results by RT-PCR

After mapping the reads to the reference genomes, viruses with genome coverage of 20% using the CLC Genomics Workbench and sequencing depth greater than 5 [7,34] were selected for further confirmation. The presence of the virus or viroid in the sample was determined by amplification of the target in RT-PCR.

Total RNA was extracted from petioles using CTAB [35]. Reverse transcription was performed with reverse transcriptase (Parstous, Mashhad, Iran) using random hexamers according to the manufacturer’s instructions. PCR was performed with the ready-to-use red PCR master mix (Amplicon, Odense, Denmark). The PCR parameters and primer sequences used for viruses/viroids are listed in Appendix A. Primers were designed based on NGS results from this study using Vector NTI v 11.0 with default parameters. The PCR product was visualized in 1% agarose with DNA Green Viewer staining; the integrity of the amplified fragments was then determined by Sanger sequencing.

### 2.5. Phylogenetic and Genetic Diversity Analysis

The genetic diversity and phylogenetic trees of the viruses and viroids potentially involved in symptom expression were assessed using the reconstructed genomes of the viruses/viroids from the sRNA libraries and the complete genomes of the corresponding sequences from different geographical regions downloaded from GenBank. The sequences were aligned using the Muscle module in MEGA v.10 [36]. The phylogenetic tree was constructed by neighbor-joining in MEGA with 1000 bootstrap replicates. Pairwise distance comparisons of genomes and color blocks representing an identity matrix were calculated using SDT v.1.3.

## 3. Results

### 3.1. Processing Raw Sequencing Data

Small RNAs isolated from the six Iranian *Vitis vinifera* cultivars were sequenced using Illumina technology. The libraries contained A4 = 14.5 × 10^6^, A5 = 15.5 × 10^6^, A6 = 15.6 × 10^6^, A7 = 16.4 × 10^6^, A8 = 15.7 × 10^6^, and A9 = 15.4 × 10^6^ unique reads (Table 1). After filtering out low-quality reads and clipping adaptors, approximately 92–99.7% of the reads remained for further analysis. The sequence data provided approximately 15.54 million reads for further downstream analysis. Finally, the recovered sequences of viruses and viroids were deposited in the Sequence Read Archive (SRA) under SRA numbers SAMN33747579-84.

### 3.2. Comparison of the Performance of Different Assemblers in the Detection and Reconstruction of Viral Genomes

Using the library-filtrated reads, the efficiency of the automated VirusDetect package and the standalone BLAST in detecting viruses was compared. The de novo assembly of the short reads into contigs was carried out using Velvet 0.7.31 [30]. Several hash lengths were tested for optimal assembly of contigs. Contigs were also subjected to standalone BLAST analysis. In VirusDetect, the filtrated reads were directly inserted into the automated pipeline for virus detection. 

Velvet de novo assembly yielded a variety of contigs ranging in length from 53 to 1464 nt. The shortest and largest contigs were assembled with k-mer = 13 and k-mer = 25, respectively.

The best hash length was k-mer = 15, which constructed 7152-19875 contigs in the range of 260–382 nucleotide lengths from the libraries (Table 2). After BLAST, it was found that the highest number of viruses and viroids were found in the contigs using k-mer = 15 (15 species), and the lowest number of identified viruses was in k-mer = 25 (2 species). GFLV, GLRaV1, AMV, and CEVd were detected in all libraries regardless of the k-mer length of the contigs (Table 2). Therefore, contigs with k-mer = 15 were de novo assembled in Velvet assembler, and BLAST analysis was used to identify viruses and viroids in the libraries.

A comparison of VirusDetect and BLAST results indicated that BLAST identified more viruses than VirusDetect. Using BLAST, 10–15 virus/viroid species were identified in the libraries, while VirusDetect could only distinguish 7–11 virus/viroid species (Table 2). 

Using the results from BLAST, the reference genome of known viruses was created, and then the performance of three programs (VirusDetect, CLC workbench v.22, and UGENE v. 41) was evaluated in read-mapping to the reference genome. The results showed that the highest number of reads were mapped to the reference genome using UGENE. The read mapping efficiency of CLC and VirusDetect was on par (Table 3 and Appendix A). Also, the highest genome length coverage was obtained with UGENE (96.95–100%), while VirusDetect and CLC workbench covered between 5.5–100% and 10–100% of the viral genomes, respectively. Regarding GFLV and GLRaV1, which have the highest number of reads in the libraries, VirusDetect, CLC workbench, and UGENE reconstructed 45.6–58.6, 48.34–70.6, and 99.6–99.7 of the genomes, respectively (Table 3, Appendix A).

### 3.3. Diversity of Virome in Iranian Infected Grapevine Cultivars

The grapevine samples used for sRNA extraction showed yellows and vein banding symptoms (Figure 1). All the viruses and viroids identified in the libraries are shown in Figure 3. Using the RNA-seq data, nine virus species belonging to seven viral genera were recognized in the libraries. Members of five families, including Betaflexiviridae (grapevine Pinot gris virus, GPGV; Grapevine virus A, GVA), Closteroviridae (grapevine leafroll-associated virus 1), Geminiviridae (grapevine red blotch virus, GRBV), Secoviridae (arabis mosaic virus, ArMV; grapevine deformation virus, GDefV; grapevine fanleaf virus, GFLV), Tymoviridae (grapevine Red Globe virus, GRGV), and the genus Virtovirus (grapevine satellite virus, GV-Sat), were identified in the virome. Four viroids (grapevine yellow speckle viroid 1, GYSVd1; hop stunt viroid, HSVd; Australian grapevine viroid, AGVd; and grapevine yellow speckle viroid 2, GYSVd2) belonging to the family Pospiviroidae were also found in the libraries. Several other viruses were also identified in the libraries, but because genome reconstruction was less than 20% of the complete genome and they were not confirmed by RT-PCR, they were considered tentative species and were not included in the Iranian grapevine virome list (Appendix A).

When considering the species abundance, GLRaV1 and GFLV were dominant in the virome. After that, GDefV, ArMV, and GRBV were abundant virus species in the samples. Moreover, HSVd and GYSVd1 were common viroids in the grapevine samples (Figure 4). 

In library A4, GLRaV1 and GFLV dominated in the virome with 53.1% and 33.74% of the viral population, respectively, whereas GDefV, ArMV, and GRBV accounted for 3.26%, 3.26%, and 1.8% of the viral population, respectively, and the other viruses occupied 3.42% of the virome (Figure 4, Table 4). In library A5, GLRaV1 and GFLV shared 54.2% and 25.56% of the virome, respectively, while GDefV, ArMV, and GRBV displayed 1.71%, 2.13%, and 1.5% of the viral population, respectively, and the other viruses had 10.5% of the population (Figure 4, Table 4). In library A6, GLRaV1, GFLV GdefV, ArMV, and GRBV represented 43.65%, 40.96%, 4.46%, 3.06%, and 0.53% of the virome, respectively, and the other viruses formed 4.89% of the population (Figure 4, Table 4). Only eight viruses were identified in library A7. GLRaV1, GFLV, GDefV, ArMV, and GRBV were the most abundant viruses in library A7, with 32.77%, 48.14%, 5.05%, 4.57%, and 0.6% frequency in the viral population. The other detected virus formed 4.57% of the population (Figure 4, Table 4). In library A8, GLRaV1 and GFLV, with 39.52% and 46.91% frequency in the viral population, respectively, were dominant species in the virome, while GDefV, ArMV, and GRBV had 2.94%, 2.33%, and 0.6% of the viral population, respectively. The other viruses formed 4.6% of the virome (Figure 4, Table 4). The highest number of identified viruses was found in library A9. GLRaV1 dominated the virome with 46.76% of the viral population. GFLV, GDefV, ArMV, and GRBV, with 29.25%, 1.92%, 3.64%, and 0.47% of the viral population, respectively, were other abundant viruses in library A9. The other thirteen identified viruses/viroids occupied 12.98% of the virome. CEVd, HSVd, and GYSVd1 were identified in all libraries, but AGVd and GYSVd2 were detected only in three libraries (A5, A6, and A9) and one library (A9), respectively (Figure 4, Table 4). 

Clustering based on virome richness revealed that the identified viruses/viroids were classified into three groups. GLRaV1 and GFLV, which formed 45.32 and 37.75 percent of the total virome, were placed in the first group. GDefV and ArMV, with an average percentage of 3.54 and 3.48 of the populations, respectively, were assigned to the second group. The other identified viruses, whose values ranged from 3.23 to 12.92% of the virome, were classified in the third group (Figure 4).

Based on the richness of the viral reads, the scrutinized libraries were divided into two clusters: libraries A4, A6, A7, and A8 were placed in the same group, and libraries A5 and A9 were grouped in another one. The number of reads of the dominant virus in the groups was slightly different (Figure 4).

### 3.4. Confirmation of Virus and Viroid Identified by HTS Data Analysis Using RT-PCR

After mapping the reads to the reference genomes, some of the viruses or viroids whose genome coverage was at least 20% in the CLC workbench with a sequencing depth greater than 5 had a chance of being found in the sample. A genome coverage threshold of 20% and a sequencing depth of 5 were used following previous work [7,37]. 

RT-PCR with specific primers was carried out to confirm the presence of AGVd [38], HSVd [38], GYSVd1 [39], GLRAV1 [40], GFLV [23], GDefV [41], ArMV [42], GRBV [43], GVA [44], AMV [45] and GarV-A [46] in grapevine tissue. Genome coverage of the selected viruses or viroids ranged from 10 to 100% of the reference genome. The results are given in Table 4.

GLRaV1 and GFLV, which had the most reads in the libraries, were detected by RT-PCR using specific primers in all samples. Their genome coverage by NGS reads ranged from 47–71% and 65–71% of the reference genome, respectively. HSVd and GYSVd1 were detected in all samples; however, AGVd was detected in only three samples (A5, A6, and A9). These results were consistent with those of HTS. Reads of these three viroids mapped to 85–100% of the reference genome. Using RT-PCR, GDefV and ArMV were detected in all libraries. The GDefV contigs in the libraries covered between 12 and 57% of the reference genome. When aligning HTS reads to the ArMV reference genome, reads mapped to 9–25.8% of the genome length (Table 3). GRBV was detected in all samples. It should be noted that only 16% of the GRBV genome was covered by the NGS reads, which is below the suggested threshold to confirm the presence of the virus in the sample. Also, GVA, with only 13.8% of the genome recovered by the NGS reads, was detected in Library A9 using RT-PCR. AMV and GarV-A were detected in two samples (A4 and A9) by NGS reads with 15.6 and 10.8% of the genome reconstruction by the NGS reads, but RT-PCR failed to detect them in the samples. It seems that the threshold required to identify a positive sample with HTS data can easily be lowered. Therefore, we suggest a genomic coverage of 14% for virus detection with HTS reads.

### 3.5. The Genetic Diversity of the Dominant Species in the Virome

Among all the viruses that were detected, the consensus sequences of GLRaV1, GFLV, and GYSVd1 were retrieved from the six scrutinized libraries and subjected to phylogenetic analysis along with the corresponding sequences in GenBank. The retrieved GLRaV1, GFLV, and GYSVd1 sequences were deposited in GenBank with accession numbers OQ849147-52, OQ627419-30, and OQ613730-36, respectively.

The nearly complete genomes of six Iranian isolates of GLRaV1 and eight complete GLRaV1 genomes from other counterparts previously deposited in GenBank were aligned and used to construct a phylogenetic tree. In the phylogeny tree, GLRaV1 isolates were put into two distinct clades with high bootstrap support (Figure 5A). GLRaV1-A4, GLRaV1-A9, GLRaV1-A8, GLRaV1-A6, and GLRaV1-A5 isolates together with mild GLRaV1 isolates from the USA (KY821089), Russia (OP727271), and Australia (AF195822) formed one clade, and GLRaV1-A7 together with GLRaV1-severe isolates from the USA (KU674797), Canada (JQ023131, NC_016509, MH807221), and France (MG925331) were in the other clade (Figure 5A). The presence of any GLRaV1 phylo-groups was not associated with grapevine cultivars. Comparative analysis revealed nucleotide (NT) identities within clade-I and Among all detected viruses, the consensus sequences of GLRaV1, GFLV, and GYSVd1 were retrieved from the six scrutinized libraries and subjected to phylogenetic analysis together with the corresponding sequences in GenBank. 

Comparative analysis revealed that nucleotide identities (NT) within group I and group II were in the range of 77.5–94.8 and 78.5–99.9, respectively. However, the mean sequence distance between the two clades was 0.16 (Figure 5A and Figure 6). The Iranian GLRaV1 isolates had a similarity of 75.9–94.8% at the nucleotide level.

For GFLV, a total of eleven RNA2 genome sequences (3656–3771 bp) were used to construct a phylogenetic tree. In the representative phylogenetic tree, Iranian GFLV isolates were divided into two branches with a mean sequence distance of 0.11 (Figure 5B and Figure 6). GFLV-A5, GFLV-A6, and GFLV-A8 isolates were in one subbranch, and GFLV-A4 and GFLV-A7 isolates were in the second subbranch of the first group, whereas GFLV-A9 was placed in a separate branch. The Iranian GFLV isolates were 84.6–93.39 identical to the corresponding sequences in GenBank. GFLV-A9 had the highest similarity of 91.75 with KJ913806 and the lowest similarity of 82.44 with MW380920 (Figure 5B and Figure 6).

A comprehensive phylogenetic analysis of GYSVd1 was carried out using seven complete GYSVd1 genomes (368–388 bp) from this study and isolates from other countries found in GenBank. All Iranian GYSVd1 isolates belonged to a group with a bootstrap support of 95, except for two sequences (GYSVd1-A9,1 and GYSVd1-A9,2) found in library A9 (Figure 5C). The Iranian GYSVd1 isolates have 93.9% to 100% identity at the nucleotide level, with little genetic distance (Figure 5C). The isolated GYSVd1 was divided into two divergent clades in the phylogenetic tree (Figure 5C). Five GYSVd1 isolates from this study (from libraries A4, A5, A6, A7, and A8) were in the same clade along with isolates from Chile (KF007310), Morocco (MH414920), Nigeria (MF576403), Thailand (KP010010), the USA (KF137564), New Zealand (HQ447058), Canada (MW732687), and Russia (ON669179). Nucleotide similarity within this group ranged from 95.6–100. GYSVd1-A9.1 and GYSVd1-A9.2 were positioned in the distant clade along with isolates from India (OL332761, OL332762), Brazil (KU880715), and China (KP993474). Members of the divergent group were 93.4–100% identical at the nucleotide level. The mean sequence distance within the main clade and the divergent clade was 0.021 and 0.031, respectively, but the mean sequence distance between the two clades was 0.029 (Figure 6).

### 3.6. The Relationship between Symptoms and the Dominant Species in the Virome

A mixed infection of GLRaV1/GFLV/GYSVd1 was detected in the virome of all samples. In libraries A4, A5, and A9, the dominant species was GLRaV1, with a concentration 1.6–2 times higher than GFLV. In libraries A7 and A8, GFLV was also the predominant virus, and its abundance was 1.45- and 2-fold higher than that of GLRaV1, respectively. In library A6, GLRaV1 and GFLV had similar concentrations (Table 4). The concentration of GYSVd1 was low in libraries A4, A5, and A8, whereas it increased 2–3.7-fold in libraries A6, A7, and A9 (Figure 7).

## 4. Discussion

In this study, the viromes of five Iranian grapevine cultivars (*V. vinifera* cv. Peykani, *V. vinifera* cv. Askari-Bidane, *V. vinifera* cv. Rezghi, *V. vinifera* cv. Sahebi, and *V. vinifera* cv. Fakhri) with high commercial importance in Iran, showing yellows or decline symptoms, were determined, and the genomes of the identified viruses/viroids were reconstructed. It can be assumed that the viruses could easily reproduce due to centuries of vegetative propagation of grapes in vineyards. The sRNAomes obtained from the six small RNA libraries from vineyards in northeastern Iran were used to perform virome analysis. The sRNA libraries of this study provided sufficient reads for the identification of viruses in the virome [47]. Previously, sRNA data were used to determine the virome of grapevine [2,48].

The sensitivity of two strategies to identify viruses from the sRNAome and the ability of aligners implemented in the three software packages to reconstruct the genome from the sRNAome of grapevines from northeastern Iran were compared. Results indicated that de novo assembly of contigs using Velvet followed Blast analysis was more efficient in virus detection than the automated VirusDetect package. The VirusDetect pipeline identified the minimum number of viruses in the libraries and was only able to detect viruses that had a huge number of reads in the library. The automated VirusDetect package has already been used to identify viruses in grapevine, soybean, sweet potato, cherry tree, and lettuce [49,50,51,52]. Also, the highest number of viruses was detected in contigs assembled with k-mer = 15 in Velvet, which is consistent with previous studies [53,54]. Therefore, when more than one assembler or k-mer was used, the chance of identifying viruses with low concentrations or those that do not have many reads in the library increased [55]. A comparison of the three aligners implemented in the CLC workbench, the VirusDetect package, and the Unipro UGENE packages to reconstruct the genomes of identified viruses from sRNAomes revealed that the Unipro UGENE aligner performed better in rebuilding the complete genome length of the target virus. Thus, examination of sRNAome data and assembled contigs by different aligners can increase the chance of reconstructing the entire viral genome [2,56].

In the present study, GLRaV1, GFLV, GDefV, ArMV, GV-Sat, GPGV, GRBV, GRGV, and GVA from the seven genera of plant viruses and GYSVd1, GYSVd2, HSVd, and AGVd from the two viroid genera of the family Pospiviroidae were identified from six libraries of five grapevine cultivars in Iran (Figure 3 and Figure 4, Table 4). ArMV [57], GdefV [58], GFLV [59], GLRaV1 [40], GPGV [60], GRGV [61], and grapevine viroids (GYSVd1, GYSVd2, HSVd, and AGVd) [38] have been previously reported from Iran; however, GRBV and GV-Sat are two new species for Iranian grapevine virome reported for the first time. These viruses comprise 1.24, 0.62, and 0.1 percent of the populations in the virome, respectively.

The probability of detecting viruses with DNA genomes in the sRNAome is low [62], but GRBV (from the genus *Grablovirus*) was identified in the samples, and more than 98% of its genome could be reconstructed. The detection and reconstruction of grapevine geminivirus A from RNAseq data were also reported [2]. 

Using the Unipro-UGENE aligner and sRNAome data, the complete genomes of the identified viruses were assembled, but the CLC workbench and VirusDetect package reconstructed only 10–87.92 and 5.5–82.74% of the genome, respectively. Complete genome reconstruction using sRNAome data has been reported previously [2,4,63,64].

The composition, abundance, and richness of viruses/viroids in the viromes were different (Figure 3 and Figure 4), but GLRaV1 with 45.82 and GFLV with 39.06% of the population were the dominant species in the virome. They were followed by GDefV and ArMV, which accounted for 3.65 and 3.27% of the virome structure, respectively (Figure 4, Table 4). Clustering of virome composition in samples (Figure 4) showed that GLRaV1 and nepoviruses, especially GFLV, were widely distributed in all samples and varieties.

Despite the high concentration of GLRaV1, leafroll symptoms were not observed in any of the samples. No clear association was found between the concentration of GLRaVs in tissue and leafroll disease (GLD) in grapevine [65]. Several reports suggest asymptomatic GLD infection with a high concentration of GLRaVs [66]; moreover, some strains of GLR-associated viruses are symptomless [67]. Moreover, with the exception of GLRaV1-A7, Iranian GLRaV1 isolates were more closely related to mild GLRaV1 isolates in the phylogenetic tree, and their distance from isolates causing severe GLD symptoms can justify the absence of leafroll symptoms.

Considering the abundance of nepovirus species such as GFLV, GDefV, and ArMV in grapevine libraries, it is likely that decline and stunting symptoms of grapevines in the surveyed vineyards are caused by infectious degeneration syndrome induced by nepoviruses [6], although mosaic and fanleaf symptoms are not observed on grapevine leaves. Previous reports also indicate that Iranian GFLV isolates represent a divergent group in the GFLV population [24,68,69], and *Vitis vinifera* cultivars in Iran do not show severe and clear symptoms of GFLV infection [70].

It also seems that cv. Pikani and cv. Fakhri are the most susceptible cultivars, and cv. Askari-bidane is the most tolerant cultivar. Most of the cv. Askari-bidane samples express moderate symptoms [70] and act as symptomless reservoir hosts, which play an important epidemiological role in the spread of the viruses.

Most of the identified viruses/viroids, e.g., GDefV, GsatV, GPGV, ArMV, GRBV, HSVd, and AGVd, can be considered as a “background” part of the virome that does not contribute to the induction of symptoms [4,71] Similar results for symptomless infection of viruses/viroids in grapevine have also been reported previously [72,73,74]. GDefV induces leaf deformation in infected grapevines, but its infection is mostly symptomless [25]. ArMV rarely generates severe mosaic symptoms on grapevine leaves [25]. GRBV generates red blotch and leaf curl on red grapevines, but symptoms on white grapevines include yellowing and severe leaf curl [75]. Also, grapevine asteroid mosaic virus (which belongs to the genus Marafivirus) causes severe yellowing symptoms in grapevines [25], but this virus was not found in Iranian grapevine virome. In addition, neither RNA-seq data nor PCR with phytoplasma-specific primers detected the phytoplasmas flavescence doree and Bois noir, which are responsible for grapevine yellows, in the samples [25]. Nevertheless, these two phytoplasmas have been reported from Iranian vineyards [76]. GYSVd1 generates yellowish speckle spots on grapevine leaves concentrated around the main veins [38].

Symptom development is a complex interplay of complex biotic (virus/host genotype) and abiotic (such as weather and climate change) factors [77]. Plants usually do not show symptoms with all viruses/viroids with which they are infected.

A mixed infection of GLRaV1/GFLV/GYSVd1 was detected in the virome of all samples. It seems that in the samples whose GLRaV1 was the predominant species in the population, yellow symptoms were observed, while in the samples in which GFLV was the predominant virus, the symptom of the vein banding strain of GFLV [78] developed. In the samples where the abundance of GYSVd1 reached almost one percent or more of the virome population, GYSVd1 enhanced the symptoms of the dominant virus, so that the most severe symptoms occurred in samples A6, A7, and A9 (Figure 1, Figure 4 and Figure 6).

It should be noted that the concentration of viroids in infected tissues is low [79]. However, it has already been shown that mutations in the genome or transmutations in the secondary structure of viroids lead to altered viroid replication, pathogenicity, or symptoms in infected plants [80,81].

Therefore, an interaction between GLRaV1 and GFLV with GYSVd1 is likely. Synergistic interaction between GFLV and GYSVd1 has been reported previously [82], but no data are available for the interaction between GLRaV1 and GYSVd1. The greatest abundance of GLRaV1 was found in libraries whose GYSVd1 concentration in the virome was low (libraries A4, A5). In contrast, except in library A9, GFLV had the greatest proportion in the virome in the libraries where the GYSVd1 population increased (libraries A6, A7, and A8) (Figure 6). It is possible that GLRaV1 and GYSVd1 have an antagonistic relationship, but further studies are needed to confirm this hypothesis and to clarify the role of GLRaV1, GFLV, and GYSVd1 in the expression of yellow symptoms in mixed infections in Iranian vineyards. In addition, the role of other background viruses/viroids should also be determined. The roles of HSVd and AGVd in generating symptoms have not yet been established. These two viroids are frequently observed in the background virome of infected grapevines [64,66,83]. HSVd has been shown to extensively reprogram genes in infected plants [84]. Further studies are needed to understand the potential role and interaction of these viroids with viruses in symptom expression.

## Figures and Tables

**Figure 1 viruses-16-00204-f001:**
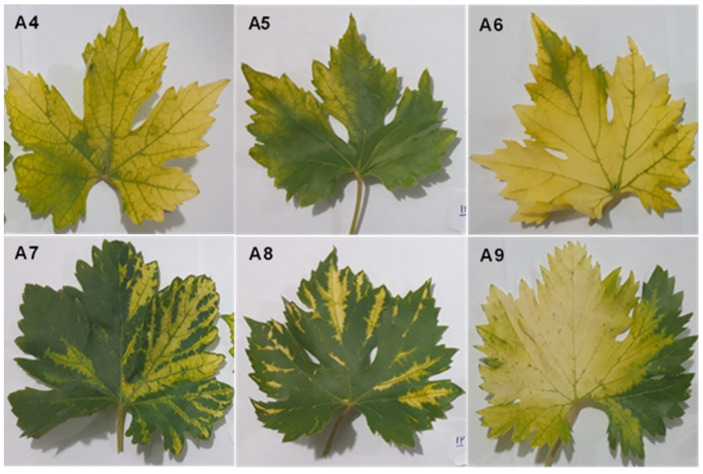
Symptoms in grapevine cultivars used for library construction. (**A4**,**A6**) Yellows in *V. vinifera* cv. Peykani; (**A5**) pale yellows in *V. vinifera* cv. Askari-Bidaneh; (**A7**) severe vein banding in *V. vinifera* cv. Rezghi; (**A8**) vein banding and sharpening of edges in *V. vinifera* cv. Sahebi; and (**A9**) yellows with severe vein banding *V. vinifera* cv. Fakhri.

**Figure 2 viruses-16-00204-f002:**
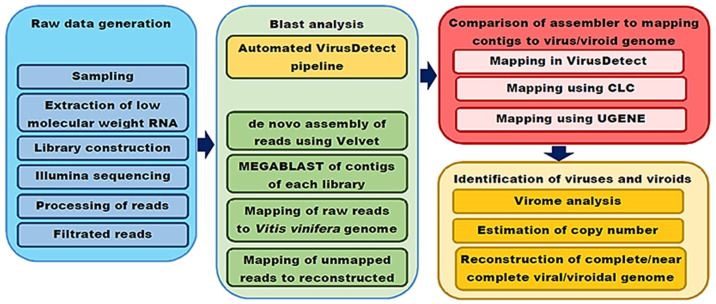
Schematic representation of the virome study of the five Iranian grapevine cultivars.

**Figure 3 viruses-16-00204-f003:**
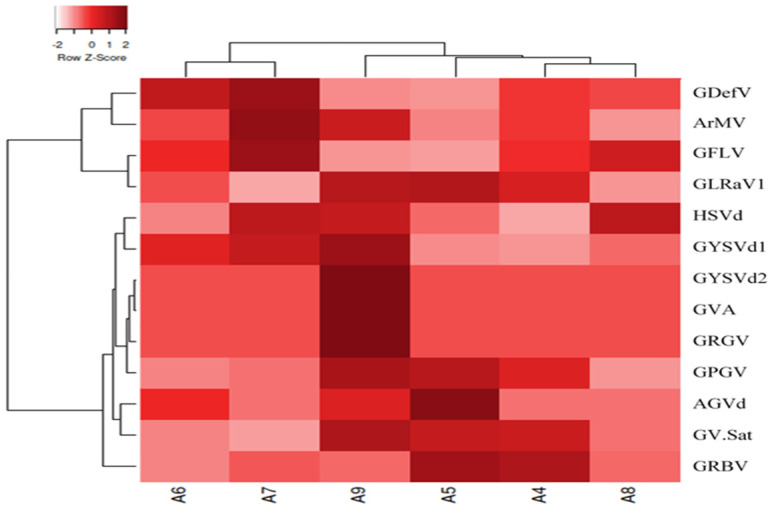
Heatmap displaying hierarchical clustering of the Iranian grapevine virome composition profiles, represented by the normalized relative abundance per grapevine sample. Heatmap color (white to dark red) displays the relative abundance of each virus across all samples. The heatmap drawn using the Euclidean dissimilarity matrix and the complete linkage distance in Heatmapper (http://www.heatmapper.ca/, accessed on 22 July 2023).

**Figure 4 viruses-16-00204-f004:**
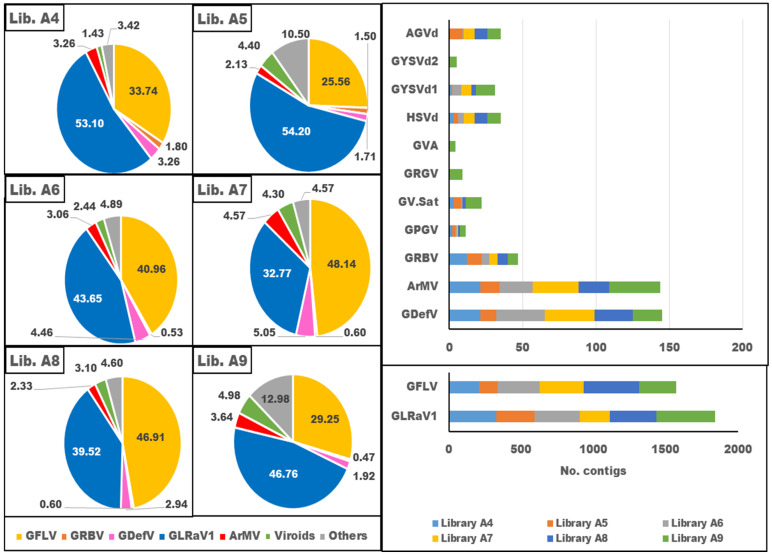
Dynamics population of virus/viroid in the virome of six grapevine sRNA libraries from Iranian vineyards. Proportion of the relative abundance of the virus/viroid in the virome of each library (**left**). Virus richness based on the number of virus/viroid specific contigs in the libraries (**right**).

**Figure 5 viruses-16-00204-f005:**
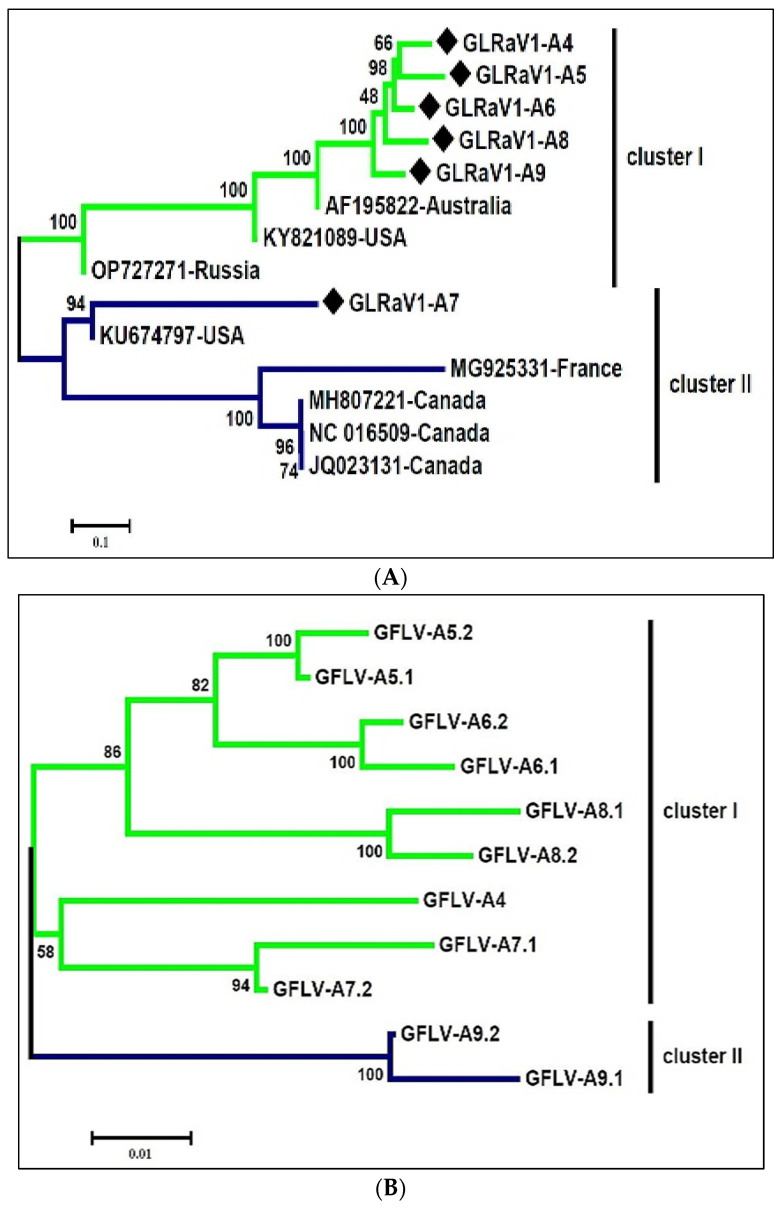
Neighbor-joining phylogenetic tree constructed from recovered consensus sequences of (**A**) Grapevine leafroll-associated virus 1 (GLRaV1), (**B**) Grapevine fanleaf virus (GFLV), and (**C**) Grapevine yellow speckle viroid 1 (GYSVd1). The tree was drawn in MEGA V.7 using the maximum composite likelihood model with 1000 bootstrap replicates. Branch length represents phylogenetic distances. Recovered isolates are shown with a ♦. Green and blue colors represent two diverse phylogenetic clades in each phylogenetic tree.

**Figure 6 viruses-16-00204-f006:**
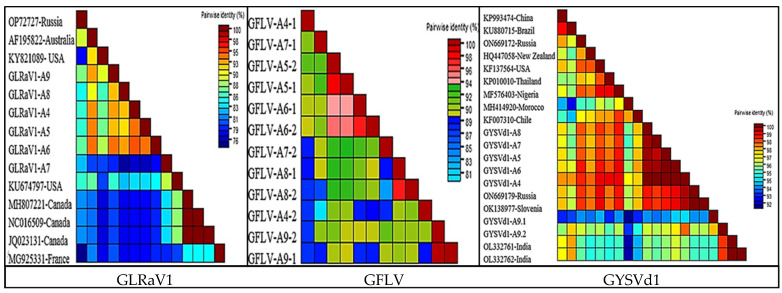
Graphical representation of pairwise nucleotide identity (with percentage identity scale) of grapevine leafroll-associated virus 1 (GLRaV1), grapevine fanleaf virus (GFLV), and grapevine yellow speckle viroid 1 (GYSVd1) with corresponding sequences from other locations. Sequence alignment using Muscle and the calculation of the identity matrix performed by SDT V.1.

**Figure 7 viruses-16-00204-f007:**
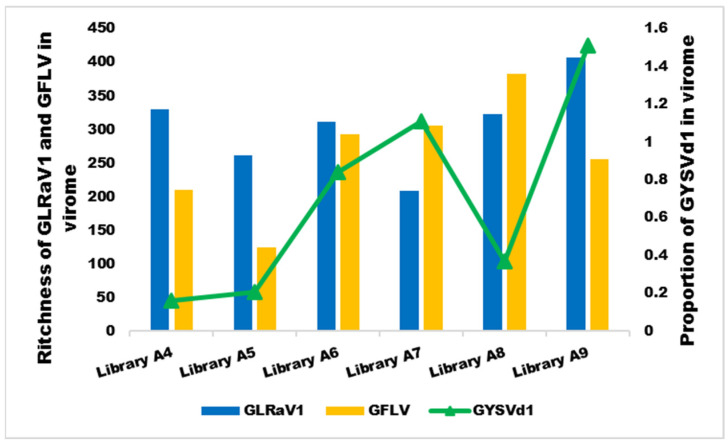
Comparison of the richness of GLRaV1 and GFLV and the relative frequency of GYSVd1 in the six Iranian grapevine libraries.

**Table 1 viruses-16-00204-t001:** A list of the grapevine libraries, including their cultivars, raw reads, number of unique reads, number of contigs, and viral/viroid associated with each library detected by NGS.

Library	Grapevine Cultivar	Raw Reads	Number of Unique Reads	No. Contigs	No. Identified Viruses	No. Identified Viroids	Viral/Viroidal Contigs	% Viral Reads/Unmapped Reads
A4	*Vitis vinifera* cv. Peykani	15,781,885	14,520,017	12,373	11	3	620	5.01
A5	*Vitis vinifera* cv. Askari-bidaneh	16,038,160	15,566,856	19,875	9	4	478	2.41
A6	*Vitis vinifera* cv. Peykani	15,775,003	15,686,878	7152	11	4	712	9.96
A7	*Vitis vinifera* cv. Rezghi	16,754,257	16,468,660	7586	8	3	631	8.32
A8	*Vitis vinifera* cv. Sahebi	15,952,556	15,766,196	9209	9	3	812	8.82
A9	*Vitis vinifera* cv. Fakhri	15,452,253	15,232,125	8595	13	5	863	10.04

**Table 2 viruses-16-00204-t002:** Number of contigs, contig length, and number of identified viruses/viroids in BLAST using contigs assembled by various hash lengths in Velvet in comparison with the results of the automated VirusDetect pipeline.

Library		A4	A5	A6	A7	A8	A9
	**unique reads**	14,520,017	15,566,856	15,686,878	16,468,660	15,766,196	15,452,253
	**hash length**						
No. contigs	K-mer: 13	16,767	17,482	10,436	11,484	11,796	11,049
	K-mer: 15	12,373	19,875	7152	7586	9209	8595
	K-mer: 17	7157	11,637	3676	4203	4744	4800
	K-mer: 19	4110	5859	1934	2101	2136	2752
	K-mer: 21	2574	3520	663	1175	931	977
	K-mer: 23	1530	2048	223	608	416	582
	K-mer: 25	1024	1287	126	368	246	304
Max contig length	k-mer: 13	87	53	77	71	55	66
	K-mer: 15	360	314	260	382	327	364
	K-mer: 17	469	578	294	410	372	351
	K-mer: 19	519	643	325	597	397	294
	K-mer: 21	678	1225	524	514	550	555
	K-mer: 23	502	827	1162	568	913	809
	K-mer: 25	514	1092	1464	1165	1172	882
No. detected virus/viroid	K-mer: 13	8	6	10	6	5	13
	K-mer: 15	11	13	12	10	11	15
	K-mer: 17	8	12	10	9	8	13
	K-mer: 19	8	12	9	9	7	12
	K-mer: 21	8	10	8	7	6	7
	K-mer: 23	8	9	6	6	5	6
	K-mer: 25	8	8	2	4	4	4
VirusDetect pipeline	No. detected virus/viroid	9	10	8	7	10	11

**Table 3 viruses-16-00204-t003:** Comparison of efficiency of three software packages in reconstruction of viral/viroidal genomes from sRNA sequences of grapevine libraries generated by high-throughput sequencing.

Virus/Viroid	Reference GenBank Accession No.	Reference Genome Size (nt)	CLC Workbench		UGENE Package		VirusDetect Pipeline	
			Genome Recovery (nt)	Genome Coverage (%)	Genome Recovery (nt)	Genome Coverage (%)	Genome Recovery (nt)	Genome Coverage (%)
Australian grapevine viroid	NC003553	370	370	100.00	370	100.00	370	100
Hop stunt viroid	NC001351	302	299	99.01	302	100.00	297	98.34
Grapevine yellow speckle viroid 2	KJ489020	363	355	97.80	363	100.00	nf	0
Grapevine satellite virus	NC021480	1060	932	87.92	1043	98.40	877	82.74
Grapevine yellow speckle viroid 1	NC001920	366	311	84.97	366	100.00	366	1
Grapevine leafroll associated virus 1	NC016509	18,659	13,177	70.62	18,596	99.66	8501	45.56
Grapevine fanleaf virus-RNA2	KU522585	3788	2306	60.88	3756	99.16	2868	75.71
Grapevine deformation virus-RNA1	NC017939	7386	3746	50.72	7332	99.27	3485	47.18
Grapevine Red Globe virus	NC030693	6863	3355	48.89	6858	99.93	614	8.95
Grapevine fanleaf virus-RNA1	KU522584	7367	3561	48.34	7341	99.65	4316	58.59
Grapevine deformation virus-RNA2	NC017938	3753	1392	37.09	3728	99.33	nf	0
Arabis mosaic virus-RNA2	NC006056	3820	1083	28.35	3762	98.48	nf	0
Arabis mosaic virus-RNA1	NC006057	7334	1891	25.78	7263	99.03	815	11.11
Grapevine Pinot gris virus	NC015782	7259	1614	22.23	7175	98.84	nf	0
Grapevine red blotch virus	NC022002	3206	513	16.00	3142	98.00	0	0
Grapevine virus A	DQ855088	7342	1013	13.80	7296	99.37	0	0

nf: not found.

**Table 4 viruses-16-00204-t004:** Description and frequency in the virome of viruses and viroids identified in five Iranian grapevine cultivars by sRNA sequencing and confirmed by PCR from composite samples of grapevine leaves collected at the commercial vineyards in the northeast of Iran.

Family	Genus	Virus /Viroid Species	Cultivar	Peykani		Askari-Bidaneh		Peykani		Rezghi		Sahebi		Fakhri	
			Library	A4		A5		A6		A7		A8		A9	
			Abbreviation	NGS	PCR	NGS	PCR	NGS	PCR	NGS	PCR	NGS	PCR	NGS	PCR
Alphaflexiviridae	Allexivirus	Garlic virus A *		0.31	-	nf	nt	0.17	-	nf	nt	nf	nt	0.28	-
Betaflexiviridae	Trichovirus	Grapevine Pinot gris virus		0.62	nt	1.91	nt	0.17	nt	0.14	nt	0.64	nt	1.05	+
	Vitivirus	Grapevine virus A		nf	nt	nf	nt	nf	nt	nf	nt	nf	nt	0.53	+
Bromoviridae	Alfamovirus	Alfalfa mosaic virus *		0.31	-	nf	-	nf	-	nf	-	nf	-	1.05	-
Closteroviridae	Ampelovirus	Grapevine leafroll associated virus 1		53.1	+	54.2	+	43.65	+	32.77	+	39.52	+	46.76	+
Geminiviridae	Grablovirus	Grapevine red blotch virus		1.8	+	1.5	+	0.53	+	0.6	+	0.6	+	0.47	+
Secoviridae	Nepovirus	Arabis mosaic virus		3.26	+	1.71	+	4.46	+	50.5	+	2.94	+	1.92	+
		Grapevine deformation virus		3.26	+	1.71	+	4.46	+	5.05	+	2.94	+	1.92	+
		Grapevine fanleaf virus		33.74	+	25.56	+	40.96	+	48.14	+	46.91	+	29.25	+
Tymoviridae	Maculavirus	Grapevine Red Globe virus		nf	nt	nf	nt	nf	nt	nf	nt	nf	nt	2.39	+
	Virtovirus	Grapevine satellite virus		nf	nt	4.77	+	0.17	nt	nf	nt	1.34	+	2.91	+
*Pospiviroidae*	Apscaviroid	Grapevine yellow speckle viroid 1		0.36	+	0.34	+	1.05	+	2.15	+	0.78	+	1.92	+
		Grapevine yellow speckle viroid 2		nf	nt	nf	nt	nf	nt	nf	nt	nf	nt	0.73	+
		Australian grapevine viroid		nf	-	3.04	+	0.69	+	nf	nt	nf	-	1.02	+
	Hostuviroid	Hop stunt viroid		1.07	+	1.02	+	0.7	+	2.15	+	2.33	+	1.31	+

nf: not found; nt: not tested. *: not supported by RT-PCR.

## Data Availability

The small RNA sequence reads in this study are available in the Sequence Read Archive (SRA) under SRA Nos. SAMN33747579-84. Also, the retrieved GLRaV1, GFLV, and GYSVd1 sequences were deposited in GenBank with accession numbers OQ849147-52, OQ627419-30, and OQ613730-36, respectively. Sequences of the PCR products were deposited in GenBank with accession numbers PP091629-59.

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
