# Peer review of "Application of High-Throughput Sequencing for Comprehensive Virome Profiling in Grapevines Shows Yellows in Iran"

_viruses, 2024, doi:10.3390/v16020204_

Round 1

Reviewer 1 Report (Previous Reviewer 1)

Comments and Suggestions for Authors

RT-PCR and sanger sequencing are necessary for new or uncommon viruses discovered by high-throughput sequencing,the authors have finished that. So I suggested that this manuscript can be accepted after minor revision.

line13: two? Results showed three approachs were used. See line 186.

Line214: GdeV? Please check it.

Author Response

We greatly appreciate the time and efforts made by the editor and referees in reviewing this manuscript.

-line13: two? Results showed three approaches were used. See line 186.

Answer: Two approaches were used for virus detection (Virusdetect vs. Blast) and three software were used for the genome mapping (CLC, VIRUS detect, UGENE)

-Line214: GdeV? Please check it.

Answer: GDefV substituted by GdeV in the manuscript

Sincerely

Reviewer 2 Report (Previous Reviewer 2)

Comments and Suggestions for Authors

The authors of the revised manuscript have diligently addressed the reviewers' comments with commendable effort. While most revisions are well-executed, there are certain areas, such as figures 4 and 5, that require further attention. For instance, the pie charts in figure 4 appear distorted, and the fonts in the phylogenetic trees within figure 5 are excessively small. To enhance readability, I recommend changing the font to Arial in figure 5 and increasing the size of the phylogenetic trees. Upon these adjustments, this manuscript will be well-prepared for publication. Congratulations on the excellent work!

Author Response

We greatly appreciate the time and efforts made by the editor and referees in reviewing this manuscript.

The pie charts in figure 4 appear distorted, and the fonts in the phylogenetic trees within figure 5 are excessively small

Answer: Figure f and 5were revized

Sincerely

This manuscript is a resubmission of an earlier submission. The following is a list of the peer review reports and author responses from that submission.

Round 1

Reviewer 1 Report

Comments and Suggestions for Authors

This manuscript described the viromes in six grapevine samples from Iran by small RNA sequencing and analysis. The authors stated that Twenty-six viruses and 16 viroids were identified from five Iranian grapevine cultivars.

The main problems:

1.      The authors described twenty-six viruses and 16 viroids were identified from the sequenced samples, however only 10 viruses were confirmed by RT-PCR. I am concerned about the presence of other viruses which were first proposed to infect grapevines. Need to confirm their exiting by Sanger sequencing, not only PCR amplification.

2.      I did not find the submitted sequences and SRA files by the ID numbers provided by the author, and these may not have been released. Still, they need to be provided as attachments so that reviewers can verify their authenticity.

3.      VirusDetect is a software skilled in finding viruses, and not software to reconstruct the genomes of viruses, how to use VirusDetect to reconstruct genomes and compare it to other software? Why not compare the three software’s ability to find some real viruses?

Other comments:

1. Remove “Deciphering the etiological black box:” in the title.

·         2. Line 16-20: Do not list viruses without Sanger sequencing validation

4.      Line 32-33: Please rephrase this sentence “As a woody plant…..viruses and viroids”

5.      Line 36-38: Are ToRSV and ArMV worldwide in grapevine?

6.      Line 50-52: Please rephrase this sentence.

7.      Line91-94: what symptoms?

8.      If these viruses are not real in these samples, these analyses in Fig 3 and 4 were unnecessary.

9.      The discussion is too long, Please make a concise discussion based on the research results

Comments on the Quality of English Language

 Moderate editing of  English language required

Author Response

We greatly appreciate the time and efforts made by the editor and referees in reviewing this manuscript.

Q1.      The authors described twenty-six viruses and 16 viroids were identified from the sequenced samples, however only 10 viruses were confirmed by RT-PCR. I am concerned about the presence of other viruses which were first proposed to infect grapevines. Need to confirm their exiting by Sanger sequencing, not only PCR amplification.

Answer: In previous studies, reconstruction of at least 20% of the complete viral genome with HTS contigs was evidence of the presence of the virus in the sample.

doi: 10.1371/journal.pone.0167966

doi: 10.1111/tpj.12294

doi: 10.3390/v10080436.

doi: 10.3389/fmicb.2022.830866

10 viruses/viroids confirmed to be present in samples using RT-PCR were reconstructed in the range of 10-100% of their genome length.

Three viruses, including GRBV, AMV, and GaVA, had less than 20% genome reconstruction with data from HTS, but were detected using specific primers in RT-PCR. ArMV and GPGV, whose genomes had less than 30% reconstruction with the HTS data, were also detected in RT-PCR.

by the way, the information on viruses whose genomes were reconstructed to less than 20% with the HTS data and were not confirmed at RT-PCR was transferred to Supplementary Tables 3 and 4.

Q2.      I did not find the submitted sequences and SRA files by the ID numbers provided by the author, and these may not have been released. Still, they need to be provided as attachments so that reviewers can verify their authenticity.

Answer: The RNA seq data have been deposited in SRA in Genbank and their accession numbers are also given in the manuscript, but the data will not be available publicly until April 2024.

Q3.      VirusDetect is a software skilled in finding viruses, and not software to reconstruct the genomes of viruses, how to use VirusDetect to reconstruct genomes and compare it to other software? Why not compare the three software’s ability to find some real viruses?

Answer: VirusDetect identifies viruses using its own internal database and uses Aligner bowtie for mapping. One of the outputs of VirusDetect is two columns in a table showing the reconstructed fragment length using the HTS data and the full length of the virus reference genome in the html and xls extension file.

Other comments

Answer: Other comments are included in the text of the manuscript

Reviewer 2 Report

Comments and Suggestions for Authors

In this study, the authors conducted small RNA sequencing to identify viruses in five Iranian grapevine cultivars. A total of 26 viruses and viroids were identified. The purpose of this study is suitable for publication; however, there are certain issues that need to be addressed before publication.

Firstly, authors should exercise caution when identifying viruses and viroids through high-throughput sequencing. The assembly of contigs was performed using the Velvet program with seven different k-mers. Based on my experience, de novo assembly with Velvet tends to generate shorter contigs compared to other assemblers. Nevertheless, for the small RNA sequencing results, the utilization of Velvet program with different k-mers is acceptable.

The major concern lies with the results presented in Table 3. The authors demonstrated virus identification using three different methods. Among these methods, de novo assembly by CLV workbench followed by BLASTX is expected to yield more reliable results compared to the UGENE package and virusdetect pipeline. The identification of garlic virus A, papaya mosaic virus, apple stem pitting virus, alfalfa mosaic virus, cucumber mosaic virus, ambrosia asymptomatic virus 2, Iranian johnsongrass mosaic virus, arabis mosaic virus, physalis rugose mosaic virus, and citrus exocortis viroid should be validated. For instance, these identifications may potentially be contaminants originating from other multiplexed libraries. It is advisable to use BLASTX search against the non-redundant protein database instead of BLASTN, as it provides more detailed information about the identified virus contigs. Many of the assembled contigs are quite small, less than 1000 bp, and may have originated from the host plants. Although RT-PCR and HTS confirmed the presence of the identified viruses or viroids, it is important to verify if these viruses or viroids can actually infect grapevines. Additionally, the authors should investigate the infection of these viruses or viroids specifically in the examined grapevines. Researchers without prior experience in HTS and bioinformatic analyses of the virome often make mistakes by misidentifying or including contaminants. Therefore, it is crucial to conduct a detailed BLASTX search for all viral contigs. Once confident in the identification, the infection of these viruses and viroids can be confirmed through RT-PCR. If the infection of these viruses or viroids in grapevines is not known, it is recommended to perform an infectivity assay specifically in grapevines.

Including all identified viral contigs as a supplementary table is of utmost importance. This will allow each reviewer to assess whether correct virus identification has been conducted. Additionally, it is essential to adhere to the International Committee on Taxonomy of Viruses (ICTV) rules when writing virus and viroid names. For instance, virus names such as "cucumber mosaic virus" should not be presented in italic font, as it is reserved for taxonomic names. The taxonomic name is a theoretical concept without physical characteristics and cannot infect plants. According to ICTV rules, taxonomic names should not be abbreviated; instead, the virus common name, such as "cucumber mosaic virus," should be written in lowercase normal font with a capital letter at the beginning of a sentence. Therefore, it should be presented as "cucumber mosaic virus (CMV)" in normal font, without italics.

Lastly, there is a discrepancy between the number of grapevines mentioned in the abstract and the materials and methods section. The abstract states the use of five Iranian grapevine cultivars, whereas the materials and methods section indicates the use of six grapevines. This inconsistency should be resolved.

I hope these edits clarify the points you wanted to convey.

Comments on the Quality of English Language

Minor editing of English language required.

Author Response

We greatly appreciate the time and efforts made by the editor and referees in reviewing this manuscript.

Firstly, authors should exercise caution when identifying viruses and viroids through high-throughput sequencing. The assembly of contigs was performed using the Velvet program with seven different k-mers. Based on my experience, de novo assembly with Velvet tends to generate shorter contigs compared to other assemblers. Nevertheless, for the small RNA sequencing results, the utilization of Velvet program with different k-mers is acceptable.

Q1.The major concern lies with the results presented in Table 3. The authors demonstrated virus identification using three different methods. Among these methods, de novo assembly by CLV workbench followed by BLASTX is expected to yield more reliable results compared to the UGENE package and virusdetect pipeline. The identification of garlic virus A, papaya mosaic virus, apple stem pitting virus, alfalfa mosaic virus, cucumber mosaic virus, ambrosia asymptomatic virus 2, Iranian johnsongrass mosaic virus, arabis mosaic virus, physalis rugose mosaic virus, and citrus exocortis viroid should be validated. For instance, these identifications may potentially be contaminants originating from other multiplexed libraries. It is advisable to use BLASTX search against the non-redundant protein database instead of BLASTN, as it provides more detailed information about the identified virus contigs.

Many of the assembled contigs are quite small, less than 1000 bp, and may have originated from the host plants. Although RT-PCR and HTS confirmed the presence of the identified viruses or viroids, it is important to verify if these viruses or viroids can actually infect grapevines. Additionally, the authors should investigate the infection of these viruses or viroids specifically in the examined grapevines. Researchers without prior experience in HTS and bioinformatic analyses of the virome often make mistakes by misidentifying or including contaminants. Therefore, it is crucial to conduct a detailed BLASTX search for all viral contigs. Once confident in the identification, the infection of these viruses and viroids can be confirmed through RT-PCR. If the infection of these viruses or viroids in grapevines is not known, it is recommended to perform an infectivity assay specifically in grapevines.

Answer: In previous studies, reconstruction of at least 20% of the complete viral genome with HTS contigs was evidence of the presence of the virus in the sample.

10 viruses/viroids confirmed to be present in samples using RT-PCR were reconstructed in the range of 10-100% of their genome length.

Three viruses, including GRBV, AMV, and GaVA, had less than 20% genome reconstruction with data from HTS, but were detected using specific primers in RT-PCR. ArMV and GPGV, whose genomes had less than 30% reconstruction with the HTS data, were also detected in RT-PCR.

by the way, the information on viruses whose genomes were reconstructed to less than 20% with the HTS data and were not confirmed at RT-PCR was transferred to Supplementary Tables 3 and 4.

All viral contigs were identified in BLASTN and BLASTX. The list of viruses in Table 1 was based on the results of BLASTX. In this context, changes were made in the text of the article in the Materials and Methods.

Q2. Including all identified viral contigs as a supplementary table is of utmost importance. This will allow each reviewer to assess whether correct virus identification has been conducted. Additionally, it is essential to adhere to the International Committee on Taxonomy of Viruses (ICTV) rules when writing virus and viroid names. For instance, virus names such as "cucumber mosaic virus" should not be presented in italic font, as it is reserved for taxonomic names. The taxonomic name is a theoretical concept without physical characteristics and cannot infect plants. According to ICTV rules, taxonomic names should not be abbreviated; instead, the virus common name, such as "cucumber mosaic virus," should be written in lowercase normal font with a capital letter at the beginning of a sentence. Therefore, it should be presented as "cucumber mosaic virus (CMV)" in normal font, without italics.

Answer: Thank you to the accuracy of the reviewer, the format of all species, genera, and families of viruses was changed based on the ICTV rules in the manuscript.

 Q3. Lastly, there is a discrepancy between the number of grapevines mentioned in the abstract and the materials and methods section. The abstract states the use of five Iranian grapevine cultivars, whereas the materials and methods section indicate the use of six grapevines. This inconsistency should be resolved.

Answer: Six RNA-seq libraries were prepared from five commercial grapevine varieties. It was corrected in the text

Other comments are included in the text of the manuscript

In the manuscript, the green mark indicates the changes requested by the reviewers, and the blue mark indicates the corrections made to the English version of the manuscript

Round 2

Reviewer 1 Report

Comments and Suggestions for Authors
  • As the manuscript showed, so many new viruses were found to infect the grapevines. We need to take this result seriously. I suggest that PCR products need to be sequenced by Sanger sequencing.

Comments on the Quality of English Language

 Moderate editing of English language required

Author Response

We greatly appreciate the time and efforts made by the editor and referees in reviewing this manuscript.

The viruses listed in Table 3 are divided into three groups.

For those for which between 80- 100% of the genome could be reconstructed using HTS data (these include AGVd, HSVd, GYSVd2, CEVd, GsatV, and GYSVd1), the presence of three viroids was verified by PCR using specific primers. In fact, 50% of the viroids present were checked by another method. These viroids have already been reported from the vineyards of Iran and the world (54 and 77 in the reference list)

The second group is the viruses whose genomes were reconstructed 30-80% (GLRaV1, GFLV, GDeV, and GRGV), of which three viruses, i.e. 75%, were analyzed by PCR. GLRaV1, GFLV and GDeV have already been reported from vineyards in Iran and around the world (23, 24, 51, and 52 in the reference list).

The third group consists of viruses whose genome was reconstructed less than 30% in the data analysis of HTS and includes ArMV, GPGV, GRBV, AMV, Gar.A, which were verified by PCR in four of the total five viruses. ArMV has already been reported from vineyards in Iran and worldwide (48 and 84 in the reference list).

An additional list of contigs obtained from each library is attached to the manuscript as a supplementary file

Reviewer 2 Report

Comments and Suggestions for Authors

In the revised manuscript, the authors have made significant revisions based on the comments from the reviewers. Most sections have been appropriately modified. However, I still have reservations regarding the presence of certain viruses and viroids in grapevines. To address this concern, it is necessary to include the sequences of all assembled viral contigs (viruses and viroids) from this study as supplementary data. By examining these assembled viral contigs, reviewers will be able to determine whether the identified viruses and viroids in this study truly originated from the examined grapevines. This clarification is essential prior to publication.

The characters in Figure 1 appear faint. I recommend changing the font type to Arial without bold and increasing the size of Figure 1.

The names of two viroids, namely GYSVd1 and GYSVd2, in Figure 3 are incomplete. Please provide a new heatmap image with the complete names.

After incorporating these minor revisions, I strongly recommend this manuscript for publication.

Author Response

We greatly appreciate the time and efforts made by the editor and referees in reviewing this manuscript.

-The revised version of figures 1 and 3 is deposited in the manuscript.

-The list of contigs of viruses listed in Table 3 for each library was prepared and attached to the manuscript as a supplementary file

Round 3

Reviewer 1 Report

Comments and Suggestions for Authors

As the manuscript showed, some viruses were reported for the first time in Iran. But the study also showedgarlic virus is reported for the first time in grapevine. I suggest the authors confirm the exiting of the new virus infecting grapevines by Sanger sequencing of PCR products.